# Adrenergic Regulation of Drp1-Driven Mitochondrial Fission in Cardiac Physio-Pathology

**DOI:** 10.3390/antiox7120195

**Published:** 2018-12-18

**Authors:** Bong Sook Jhun, Jin O-Uchi, Stephanie M. Adaniya, Michael W. Cypress, Yisang Yoon

**Affiliations:** 1Lillehei Heart Institute, Cardiovascular Division, Department of Medicine, University of Minnesota, Minneapolis, MN 55455, USA; stephanie_adaniya@brown.edu (S.M.A.); mcypress@umn.edu (M.W.C.); 2Cardiovascular Research Center, Rhode Island Hospital, Providence, RI 02903, USA; 3Department of Medicine, Division of Cardiology, the Alpert Medical School of Brown University, Providence, RI 02903, USA; 4Department of Physiology, Medical College of Georgia, Augusta University, Augusta, GA 30912, USA; yyoon@augusta.edu

**Keywords:** adrenoceptor, Ca^2+^/calmodulin-dependent protein kinase II (CaMKII), protein kinase A (PKA), protein kinase D (PKD), calcineurin, GTPase, mitochondrial permeability transition pore, apoptosis, phosphorylation

## Abstract

Abnormal mitochondrial morphology, especially fragmented mitochondria, and mitochondrial dysfunction are hallmarks of a variety of human diseases including heart failure (HF). Although emerging evidence suggests a link between mitochondrial fragmentation and cardiac dysfunction, it is still not well described which cardiac signaling pathway regulates mitochondrial morphology and function under pathophysiological conditions such as HF. Mitochondria change their shape and location via the activity of mitochondrial fission and fusion proteins. This mechanism is suggested as an important modulator for mitochondrial and cellular functions including bioenergetics, reactive oxygen species (ROS) generation, spatiotemporal dynamics of Ca^2+^ signaling, cell growth, and death in the mammalian cell- and tissue-specific manners. Recent reports show that a mitochondrial fission protein, dynamin-like/related protein 1 (DLP1/Drp1), is post-translationally modified via cell signaling pathways, which control its subcellular localization, stability, and activity in cardiomyocytes/heart. In this review, we summarize the possible molecular mechanisms for causing post-translational modifications (PTMs) of DLP1/Drp1 in cardiomyocytes, and further discuss how these PTMs of DLP1/Drp1 mediate abnormal mitochondrial morphology and mitochondrial dysfunction under adrenergic signaling activation that contributes to the development and progression of HF.

## 1. Introduction

Mitochondria are essential eukaryotic organelles that generate the energy necessary for a myriad of cellular processes (see reviews [1,2,3,4,5]). Cellular distribution, organization, and the shape of mitochondria vary greatly in different mammalian cells/tissues, which is presumed to be a result of adaptation for specific functions of highly differentiated organ systems in multi-cellular organisms [1,6,7]. It is widely reported using cell culture systems that mitochondria can change their shape and location frequently, collectively termed “mitochondrial dynamics”, which is suggested as one of the most important modulators for bioenergetics, reactive oxygen species (ROS) generation, spatiotemporal dynamics of Ca^2+^ signaling, cell growth, and death in cell- and tissue-specific manners [8,9,10,11]. The most well-characterized processes for changes in mitochondrial shape are fission and fusion, which are regulated by the members of the dynamin family of large GTPases ubiquitously expressed in the various cells/tissues; Dynamin-like/related protein 1 (DLP1/Drp1) drive mitochondrial fission, whereas mitofusin 1 and 2 (Mfn1 and Mfn2), and optic atrophy 1 (OPA1) mediate fusion of the outer and inner mitochondrial membranes, respectively [12,13,14].

The mitochondrial dynamics of excitable cells such as cardiomyocytes (CMs) have attracted attention among researchers due to their cellular energetic demands as well as their abundant expression of mitochondrial fission/fusion proteins [15,16]. In adult CMs, the globular-shaped mitochondria are densely packed into the intermyofibrillar spaces with one or more mitochondria for each sarcomere, which likely restricts mitochondrial dynamics [15,16,17,18]. Therefore, mitochondrial dynamics in striated muscles (i.e., cardiac and skeletal muscles) as well as their physiological relevance to mitochondrial and cellular function have long been under debate. By using a photoactivable green fluorescent protein (GFP) in the mitochondrial matrix both in isolated cells and in vivo, it has been reported that mitochondria form local networks and undergo fusion events to share mitochondrial matrix content to neighboring mitochondria in both adult skeletal and cardiac muscles [19,20,21]. However, Hajnoczky’s group further demonstrated that these fusion events (or mitochondrial networks/interactions) were significantly slower (≅3–5 min), less frequent, and more stable compared to non-differentiated myoblasts or neonatal cells [19,20]. They also found that fusion events are dependent on the activity of mitochondrial fusion proteins, OPA1 and Mfn1 [20]. Indeed, Balaban’s group also used three-dimensional (3D) electron microscopy (EM) to show that mitochondria in both skeletal and cardiac muscles form highly connected networks and communicate through junctions, although they did not precisely demonstrate whether there is continuity in the matrix all along the conducting elements [22,23,24]. Thus, this emerging evidence suggests that the muscle mitochondria possess at least slow fusion machinery activity under physiological conditions, which maintains limited mitochondrial networks inside the myofibrils. However, since the experimental techniques for detecting mitochondrial fission events in live myofibrils are still being developed [25], the frequency and speed of fission events in striated muscles including adult CMs remain unclear. Moreover, it is largely unknown whether mitochondrial fission/fusion events influence the beat-to-beat-based regulation of excitation–contraction (E–C) coupling in CMs. 

In the heart (especially in the ventricles), cardiac mitochondria occupy over 30% of the cell volume [6,26] and are usually classified into three groups according to their location: intermyofibrillar mitochondria (IFM), subsarcolemmal mitochondria (SSM) and perinuclear mitochondria (PNM) [7,15]. Importantly, abnormal mitochondrial morphologies concomitant with mitochondrial dysfunction in the heart are frequently observed in both human patients and animal models of cardiac diseases. For instance, transmitted EM (TEM) images from human ventricle samples of cardiomyopathy and heart failure (HF) frequently show the misalignment of IFM lying along the sarcomeres with smaller, rounder mitochondria compared to non-failing hearts [27,28,29,30]. In line with these observations in human hearts, animal models of HF through hypertrophy and ischemia exhibit similar changes in IFM morphology [31,32,33,34,35,36,37]. On the other hand, in animal hearts after acute ischemia, cardiac mitochondria tend to maintain their morphology and integrity, but the matrix density of IFM and SSM are significantly decreased, as seen in EM images [38,39]. This is mainly caused by opening of the mitochondrial permeability transition pore (mPTP) [40]. These observations indicate that mitochondrial fragmentation frequently occurs under chronic pathological conditions such as HF in the heart, possibly due to increased mitochondrial fission activity [41] and/or the inhibition of the fusion activity [20]. Notably, fragmented mitochondria are frequently observed in a variety of human diseases including neurodegenerative diseases, diabetes, and cancer in addition to cardiac diseases [42,43]. Although the evidence clearly suggests a link between mitochondrial fragmentation and cardiac pathology, several important questions remain:
(1)Which cardiac signaling pathways regulate mitochondrial morphology and function under pathophysiological conditions?(2)How do the cardiac signaling pathways modulate the activities of mitochondrial fission and fusion proteins?(3)Do abnormal morphologies of cardiac mitochondria contribute to the development of cardiac pathology or are they just an adaptation/maladaptation to pathological changes in the heart?

Since the early 1990s, sympathetic nervous system activation and its downstream signaling in CMs (i.e., adrenoceptor (AR) signaling) has been well recognized as a main factor causing or exacerbating cardiac pathology, especially HF [44,45,46,47,48,49], in addition to its role as the primary regulator of E–C coupling under physiological conditions [50,51] (Figure 1). Moreover, growing evidence indicates that post-translational modifications (PTMs) of mitochondrial fission and fusion proteins play a role in regulating mitochondrial dynamics in various cell types/tissues [52,53,54,55]. Recently, new studies have reported novel signaling mechanisms located downstream of AR (e.g., kinases and phosphatases), which lead to PTMs of a mitochondrial fission protein DLP1 and regulate mitochondrial and cellular function in CMs [56,57,58,59,60]. Therefore, in this review, we summarize molecular mechanisms underlying PTMs of DLP1 in regulating mitochondrial fission under AR signaling, and further discuss whether PTMs of DLP1 by AR signaling pathways mediate mitochondrial and cellular injury in HF.

## 2. Overview: Adrenergic Regulation of Mitochondrial Functions in Cardiomyocytes

In adult CMs (in this review, we refer mostly to adult ventricular myocytes), several subtypes of ARs including β_1_-, β_2_-, β_3_-, and α_1_-ARs [45,61,62,63,64] are expressed with β_1_-AR being the most abundant [49]. As mentioned above, AR signaling is an important regulator for physiological E–C coupling on a beat-to-beat basis. Catecholamines (i.e., epinephrine and norepinephrine) act on these ARs in CMs to regulate heart inotropy, chronotropy, and growth [45,49,50,61,62,63,64,65,66] via AR subtype-specific downstream pathways [45,61,65,67]. For instance, β-AR signaling increases the transient rise of cytosolic Ca^2+^ concentration ([Ca^2+^]_c_) (i.e., Ca^2+^ transient: CaT) about 1.5–2-fold at each beat, which triggers increased cardiac contraction [50,51]. This mechanism has been well-described in terms of the acute PTMs (mainly phosphorylation) of the major Ca^2+^ handling proteins such as voltage-gated L-type Ca^2+^ channel (VLCC), ryanodine receptor type 2 (RyR2), and phospholamban. The stimulation of α_1_-AR also increases CaT, although much less than that of β-AR stimulation (≅10–15%), but also contributes to positive inotropic effects in the heart. This difference in the contribution of β-ARs and α_1_-AR to E–C coupling is mainly due to the difference in downstream signaling activated under each AR. ARs are part of the large family of G protein-coupled receptors (GPCRs) and AR subtypes couple to different G proteins (e.g., β_1_- and α_1_-AR interact mainly with G_s_ and G_q_, respectively), thus activating distinct signaling pathways [45,46] (Figure 1), though cross-talk signaling between α_1_- or β-ARs may also exist [45,68,69,70]. Under acute AR stimulation (e.g., “fight-or-flight” responses), heart rate and contractile force generated by CMs increase, indicating that cardiac mitochondria increase their ATP production temporarily for this additional energy demand. Though further investigations are required, possible mechanisms for matching the energy demand to supply under acute AR stimulation include: (1)β-AR signaling dramatically increases CaT, promoting the acceleration of Ca^2+^ uptake from the cytosol to the mitochondrial matrix via mitochondrial Ca^2+^ uniporter (MCU), thus increasing mitochondrial matrix Ca^2+^ ([Ca^2+^]_m_) and consequently activating the tricarboxylic acid (TCA) cycle and oxidative phosphorylation (OXPHOS) to enhance ATP production [71,72].(2)α_1_-AR stimulation increases the phosphorylation of MCU, which additionally enhances Ca^2+^ uptake, maintaining the higher [Ca^2+^]_m_ [73,74] (see Figure 1).

Importantly, [Ca^2+^]_m_ elevation by prolonged β-AR or α_1_-AR stimulation (e.g., ≥15 min) also increases mitochondrial ROS (mROS) generation, which triggers mPTP opening or arrhythmogenic [Ca^2+^]_c_ oscillation [74,75]. Sheu and colleagues showed that increased [Ca^2+^]_m_ evoked by [Ca^2+^]_c_ elevation using an inhibitor of sarco/endoplasmic reticulum Ca^2+^-ATPase (SERCA) thapsigargin can promote DLP1 translocation to the mitochondria, followed by mitochondria fragmentation and increased mROS in cultured CMs [41]. However, there are several remaining questions to be resolved:
(1)Does DLP1 translocation occur under physiological [Ca^2+^]_c_ elevation, such as increased CaT under acute β-AR stimulation?(2)Does DLP1 translocation occur via increased [Ca^2+^]_m_ and/or [Ca^2+^]_c_?(3)Are mitochondrial dynamics, especially fission, involved in the process for energy matching and/or mROS production under AR stimulation (see also Section 4)?

In addition to acute and subacute effects of AR signaling, it is well-established that chronic AR stimulation promotes pathological remodeling, leading to cardiac hypertrophy and/or HF [49,76]. Since α_1_-AR density in the heart is considerably lower than that of β-ARs (≅20% in human myocardium), the contribution of α_1_-ARs to AR-mediated cardiac hypertrophy may be less prominent than that of β-ARs in vivo [61]. However, another important observation is that β-AR signaling is downregulated via β-AR internalization under pathophysiological conditions including HF (see also Figure 1), whereas α_1_-AR expression at the plasma membrane has been shown to be unchanged [62,67,77]. Therefore, as the relative proportion of α_1_-AR increases, α_1_-AR stimulation has a more important role exacerbating pathological conditions of myocardium such as HF. Indeed, in addition to β-AR pathways, signaling pathways of cardiac G_q_-protein-coupled receptors (G_q_PCRs)—including α_1_-ARs, angiotensin II receptors, and endothelin I receptors—are critical for the development and progression of HF [78,79]. It should be noted that experimentally-induced chronic AR stimulation causes mitochondrial fragmentation in isolated CMs and in vivo animal hearts [56,60,80,81,82,83,84]. Though it is still unclear whether the alteration in the stoichiometric ratio of fission/fusion proteins creates an imbalance between fusion and fission in HF ([34,35,85]), these observations have introduced the idea that PTMs of mitochondrial fission/fusion proteins by AR signaling may be at least part of the mechanism for promoting mitochondrial fragmentation in the CMs during HF. Indeed, there are few publications showing the PTMs of mitochondrial fusion protein and their functional consequences (i.e., inhibition of fusion) in the CMs [86,87,88]. In this review, we specifically focus on the PTMs of mitochondrial fission protein DLP1 and discuss the possible molecular mechanisms of enhanced mitochondrial fission under chronic AR stimulation (see following Section 3 and Section 4). 

## 3. DLP1-Mediated Mitochondrial Fission in Non-Cardiac and Cardiac Cells

The major proteins involved in mammalian mitochondrial fission are a soluble cytosolic protein DLP1 (also called Drp1) and its receptor mitochondrial fission factor (Mff) which is anchored at the outer mitochondrial membrane (OMM) (see reviews [89,90,91,92]). In addition, DLP1 is known to associate with various membrane structures in the cell including peroxisomes [93], lysosomes, late endosomes and the plasma membrane [94]. DLP1 possesses GTPase activity that provides DLP1 with membrane remodeling activity [95,96] after associating with the membrane structures such as OMM or the peroxisomal membrane, thus promoting fission of mitochondria or peroxisomes, respectively [93,97,98]. The domain structure of DLP1 includes the N-terminal GTPase domain, middle domain, variable domain (VD, also known as B-insert), and the C-terminal GTPase effector domain (GED) [99,100,101] (Figure 2). 

Based on the reports of crystal structure of DLP1, this protein has a globular head and stalk structures, similar to conventional dynamin [100]; the GTPase domain forms the head, and the stalk is composed of a middle domain and GED connected by a loop formed by B-insert; hence, the B-insert is located at the tip of the stalk, at the opposite end of the GTPase head domain. Importantly, the B-insert loop contains the two well-known serine (Ser or S) phosphorylation sites (S616 and S637) that determine mitochondrial translocation of DLP1 (Figure 2), suggesting that phosphorylation-mediated conformational changes may regulate DLP1 recruitment to the OMM [100,103,104].

A DLP receptor, Mff is a C-terminally anchored protein located at the OMM [105,106] and the peroxisomal membrane [107], whereas mitochondrial dynamics proteins of 49 and 51kDa, termed MiD49 and MiD51, respectively, are N-terminally anchored proteins at the OMM [108,109]. While both Mff and MiD49/51 are DLP1 receptors, the relative contribution of each receptor for the overall fission activity in various cell types/tissues is still controversial [110,111]. In the heart, Mff is likely the primary receptor at the OMM, since the loss of Mff in vivo causes a significant decrease in mitochondrial fission and more severe cardiomyopathy [112]. Based on recent publications [104,113,114,115], a model of how DLP1-Mff interaction at OMM contributes to fission is proposed as follows:
(1)B-insert structure of DLP1 normally blocks the Mff-binding sites in DLP1.(2)B-insert binds to cardiolipin at the OMM, which allows the exposure of the Mff-binding site of DLP1 for DLP1-Mff interaction.(3)Binding of DLP1 dimers to Mff allows DLP1 assemblies to form functionally active DLP1 oligomers on the OMM in the form of a helical ring. GTP-induced constriction of the DLP1 helical ring plays a crucial role in mitochondrial membrane fission.

Since B-insert contains sites for phosphorylation, O-GlcNAcylation, and SUMOylation [56,58], PTMs of DLP1 may induce a conformational change of B-insert that regulate interactions between DLP1 and Mff on the membrane undergoing fission. However, further studies are required to delineate the detailed molecular mechanism of how the interactions between DLP1 and Mff are regulated at the OMM to provide the force for fission. Moreover, in addition to mitochondrial fission, DLP1-Mff interactions are also critical for peroxisomal fission. Considering that fatty acid metabolism is the primary energy source of the heart, the mitochondria and peroxisomes are the two main sites of β-oxidation of fatty acids in CMs [116], and peroxisome proliferator-activated receptors (PPARs) has been shown to be important pathways for mitochondrial protection and biogenesis in the heart [117,118,119], it is reasonable to surmise that the regulation of peroxisome morphology by DLP/Mff may have important roles for modulating cardiac energy metabolism and/or ROS production. However, there are currently only a few studies assessing peroxisome morphology in adult CMs. Further studies are required for precisely understanding the role of DLP/Mff on peroxisome function, morphology and the consequent role in regulating energy metabolism in the heart.

Multiple PTMs of DLP1 have been detected by mass spectrometry-based proteomics as shown in the online open data base (e.g., PhosphoSitePlus [102]) (Figure 2). Some of the PTM sites in DLP1 have been further investigated, and their physiological and pathological relevance to mitochondrial and cellular functions have been assessed in various cell types including CMs (see also reviews from others [52,55,89]). These include (1) phosphorylation, (2) ubiquitination, (3) SUMOylation, (4) S-nitrosylation, and (5) O-linked-N-acetyl-glucosamine glycosylation (O-GlcNAcylation), which can change functional properties of DLP1 such as subcellular localization, protein–protein interactions (e.g., DLP1-Mff binding), protein stability, and GTPase activity. Figure 2 and Table 1 summarize the PTM sites within the DLP1 structure, the mediating molecules (e.g., kinases and phosphatases), and their effects reported in non-cardiac cells and CMs.

Among these phosphorylation sites on DLP1, S616 and S637 are well reported (Figure 2). Amino acid sequences surrounding S616 matches the consensus motif for Cyclin-dependent kinase (CDK1/cyclin B) [131,132,133,134] and extracellular signal–regulated kinase 1/2 (ERK1/2) [103,136,137,138], and this site was shown to be phosphorylated by these kinases. In addition, protein kinase Cδ (PKCδ) and Ca^2+^/calmodulin-dependent protein kinase II (CaMKII) have been shown to be upstream kinases for DLP1 phosphorylation at S616, although the sequences around this site do not match the consensus motifs of these kinases [57,133,139,140]. Another Ser phosphorylation site S637 is reported as a site for protein kinase A (PKA) [58,145,146,147], Ca^2+^/calmodulin-dependent protein kinase Iα (CaMKIα) [148], protein kinase D (PKD) [56], Ca^2+^-dependent phosphatase calcineurin (CaN) [60,144,151], and protein phosphatase 2 (PP2A) [153,154]. Importantly, S637 is located within a consensus motif for Rho-associated protein kinase (ROCK), and its phosphorylation was detected in endothelial cells [149]. However, Brand and his colleagues showed that in neonatal CMs, ROCK cannot phosphorylate S637, but can phosphorylate S616 [141].

S616 phosphorylation consistently has resulted in enhanced mitochondrial fission in various cell types tested by multiple labs. However, the functional consequences of S637 phosphorylation for regulating mitochondrial morphology is still highly controversial [167]. For instance, various reports on the overexpression of phospho- or dephospho-mimetic mutants of DLP1 (S637D and S637A, respectively) have resulted in conflicting outcomes for mitochondrial morphology: S637D introduction promotes mitochondrial elongation [58,144,145], or fragmentation [148]; and S637A promotes fragmentation [58,144], elongation [148], or no significant changes in mitochondrial morphology [56,145] (Table 1). Moreover, another PTM of DLP1, O-GlcNAcylation at threonine (Thr or T) 585 and 586, has been shown to lead to dephosphorylation of DLP1 at S637, followed by DLP1 translocation to the OMM and mitochondrial fragmentation under hyperglycemic conditions in neonatal CMs [129] (Table 1). However, we recently reported that PKD-dependent phosphorylation of S637 promotes DLP1 translocation to the OMM and mitochondrial fragmentation under α_1_-AR stimulation in neonatal CMs [56]. The differing and conflicting outcomes of DLP1-S637 phosphorylation observed by different groups shown above may be partly due to:(1)Different cell types used for the overexpression of DLP1 mutants which possess distinct mitochondrial shapes/networks in resting conditions (i.e., highly elongated and interconnected, or mixture of small and tubular mitochondria).(2)Involvement of different upstream molecules (i.e., kinases and phosphatases) listed above.(3)Co-existence of additional PTMs within DLP1 (or in other fission/fusion proteins) caused by the different upstream molecules in addition to S637 phosphorylation.

These discrepancies will be discussed further in Section 4, specifically in the case of CMs. DLP1 has also been shown to be ubiquitinated by E3 ubiquitin ligases including MARCH5 (also known as *MITOL*) [160,161,162,163] and Parkin [164,165] (Table 1). Initial studies for MARCH5-DLP1 interactions showed DLP1 ubiquitination and consequent degradation [160,161], which may be a direction towards decreased fission activity. However, later studies suggest that DLP1 ubiquitination by MARCH5 instead promotes mitochondrial fission, possibly by mediating the recruitment of DLP1 at specific fission sites [162,163]. On the other hand, the consensus understanding for Parkin-mediated DLP1 ubiquitination is that this signal induces DLP1 degradation in a proteasome-dependent pathway [164,165]. 

SUMOylation is a PTM that promotes the addition of small ubiquitin-like modifier [SUMO] proteins to multiple lysine (Lys or K) residues of various proteins including DLP1 [122,168] (Figure 2 and Table 1). The SUMOylation of DLP1 is driven by a mitochondrial-anchored SUMO E3 ligase, mitochondrial-anchored protein ligase (MAPL), that stabilizes the oligomeric form of DLP1 on the OMM to increase mitochondrial fission during apoptosis [123,124]. On the other hand, Sentrin/SUMO-specific protease 5 (SENP5) participates in the deSUMOylation of DLP1, which reduces mitochondrial fission [125]. 

S-nitrosylation of DLP1 has been reported in neuronal cells but may need further investigation to confirm its functional relevance due to the limited number of publications from only a few groups to date (Figure 2 and Table 1). For instance, nitric oxide (NO)-mediated S-nitrosylation of DLP1 at cysteine (Cys or C) 644 was proposed to increase mitochondrial fission in neurons, and contribute to the pathologies of neurodegenerative diseases including Alzheimer’s and Huntington’s diseases especially from Lipton’s group [156,157] (see also their reviews [169,170]). However, other groups’ studies indicate that S-nitrosylation of DLP1 itself does not change DLP1 activity [155], but rather NO or S-nitrosylation of DLP1 mediates increased S616 phosphorylation to promote fission [155,158].

As listed above, in addition to phosphorylation and O-GlcNAcylation, several other PTMs of DLP1 (ubiquitination, SUMOylation, and S-nitrosylation) were reported in cultured cell lines or non-cardiac primary cells. However, these PTMs have not yet been established as having physiological and/or pathological roles in the CMs. Therefore, in the following section, we will briefly discuss the possibility that these PTMs participate in signaling pathways that promote or exacerbate HF.

Finally, it should be mentioned that several reports show the PTMs of DLP1 receptors (adenosine monophosphate-activated protein kinase [AMPK]-dependent phosphorylation of Mff [171,172] and MARCH5-dependent ubiquitination of MiD49 [173]) by cellular signaling that may modulate mitochondrial fission activity. Though these findings are also crucial in understanding the whole picture of regulation of mitochondrial fission activity by cardiac signaling pathways, we mainly focus on PTMs of DLP1 in this review due to the limited number of reports related to PTMs of DLP1 receptors compared to those of DLP1.

## 4. Molecular Mechanisms Underlying the Modulation of DLP1 Function by Adrenergic Signaling in Cardiomyocytes

### 4.1. Suitable Models for Investigating Post-Translational Modifications of DLP1 and Their Roles in Cardiomyocytes

Historically, DLP1 research has thrived off knowledge obtained from cultured cell lines and/or non-excitable cells (Table 1). Moreover, the physiological and pathological importance of DLP1 function in vivo has emerged from recent clinical reports showing that individuals who possess a DLP1 mutation exhibit neurological disorders [174,175,176,177,178]. Although a DLP1 mutation (DLP1-C452F) causing cardiomyopathy was reported in mice [179,180], cardiac disorders in patients who possess DLP1 mutations have not been yet documented. These prior studies lead us to the idea that other types of DLP1 modifications in CMs are likely to have more pivotal roles for driving cardiac dysfunctions in vivo compared to its mutations. In this context, many laboratories have started investigating the physiological and pathological roles of DLP1 and its PTMs, specifically in the adult CMs (especially, adult ventricular myocytes)/hearts. However, there are multiple technical challenges for using adult CMs in DLP1 research that needs to be overcome. These include:(1)Difficulty in observing mitochondrial morphology/dynamics (especially fission events) in adult CMs under light/confocal microscopy [19,20] (See also Section 1)(2)Difficulty in maintaining the specific cellular membrane integrity (i.e., T-tubules) in cultured adult CMs [11,57,74](3)Limitation for gene manipulations in isolated CMs without culturing certain periods (e.g., 24–48 h) [11,57,74]

For point (1), a combination of the conventional TEM and state-of-the-art 3D EM [22,23,24] provides us with some clues (see also Introduction), although live cell imaging is still a challenge. Mainly due to the points of (2) and (3), it is still a reasonable idea to extract information of cardiac-specific DLP1 function from various “culture cell models for adult CMs” that possess a similar structure of signaling pathways, even though their mitochondrial morphologies/dynamics at resting conditions differ from those of adult CMs [181]. For instance, the following cell types are commonly used in addition to cultured adult CMs: HL-1 mouse atrial CM cell line [133], H9c2 cardiac myoblasts [56,57,182], primary neonatal CMs [56,141,150], and human induced pluripotent stem cells (iPSCs)-derived CMs [183] (see also the notes for Table 1). Therefore, we introduce the data from culture cell models mentioned above in addition to adult CMs. The use of putative DLP1 inhibitors, including Mdivi-1, has been widely applied to reduce the GTPase activity of DLP1 in cells and in vivo, which frequently shows protective effects from mitochondrial injury and cell death (see review [184]). However, the effects of these beneficial effects are likely via non-specific inhibition of complex I by these drugs [185], and thus the data obtained by the use of Mdivi-1 may not be suitable for understanding the role of DLP1 function as well as PTMs of DLP1. Therefore, to precisely understand the role of PTMs of DLP1 under AR stimulation in CMs, we develop our discussion here based on the accumulating data using the mutants of DLP1 that mimic or diminish PTMs of DLP1 as well as the experimental results from the genetic modification of upstream signaling pathways, rather than the reports using the pharmacological inhibition of DLP1 activity and upstream signaling. 

### 4.2. Overview of DLP1 Phosphorylation by Adrenergic Signaling in Cardiomyocytes

It is widely accepted that chronic AR stimulation (either the stimulation of β-AR alone [81,82,83,84], α_1_-AR alone [56,80], or both [60], which commonly occurs under HF, contributes to mitochondrial fragmentation and dysfunctions in CMs and/or in vivo animal hearts. In addition, emerging evidence shows that chronic G_q_PCR stimulation including α_1_-AR also causes mitochondrial fragmentation and dysfunctions [186,187,188]. These observations lead to the idea that prolonged stimulation of G_s_/G_q_-protein signaling promotes mitochondrial fission in CMs. Several signaling molecules under the cascades of AR subtypes and G_s_/G_q_ proteins (see Figure 1) were found as a possible candidate to modulate DLP1 via its PTMs in CMs (Figure 3A) (see also Section 3). As shown in Figure 3B and Table 1, most accumulated records of publications from adult CMs, cultured CM-model cells (see Section 4.1), and cardiac tissues concern phosphorylation at S616 and S637. Therefore, next we will summarize the possible molecular mechanisms for phosphorylation at S616 and S637 in DLP1 by each AR-subtype signaling pathway and discuss the physiological and pathological relevance of each phosphorylation to mitochondrial morphology and function.

### 4.3. Phosphorylation of DLP1 at S616 by β-Adrenoceptor Signaling

Wang’s group showed that S616 is phosphorylated in isolated adult CMs by chronic treatment (over 12 h) of a β-AR agonist isoproterenol (Iso) [57]. Moreover, 2-week continuous infusion of Iso in vivo also promotes significant S616 phosphorylation in the heart. They also confirmed that (1) β-AR-mediated S616 phosphorylation is via CaMKII in CMs, and (2) S616 phosphorylation promotes DLP1 translocation to the OMM and increases mitochondrial flashes (i.e., the rate of transient mPTP opening [189]) (Figure 3B), although the sequences around this site do not match the consensus motif of CaMKII [57]. 

### 4.4. S616 Phosphorylation of DLP1 by α_1_-Adrenoceptor Signaling

Several upstream kinases have been reported both in non-cardiac and cardiac cells that may be activated in CMs under α_1_-AR stimulation, which include CaMKII, PKCδ, and ERK1/2 (See Figure 1, Figure 3A, and Table 1). RhoA/ROCK [141] can also be activated under α_1_-AR-G_q_ by an unknown mechanism, but this signaling is not efficacious compared to G_12/13_ protein-coupled receptor stimulation such as sphingosine 1-phosphatase (S_1_P) receptors [190]. All three potential candidate kinases listed above are known to be acutely activated by α_1_-AR stimulation (Figure 1 and Figure 3), but we have shown that 30-min treatment of an α_1_-AR agonist phenylephrine (Phe) did not promote S616 phosphorylation in DLP1 in H9c2 cells [56]. On the other hand, we also found that significant increases in both S616 and S637 phosphorylation in ventricular tissues from transgenic mice with the overexpression of a constitutively active G_q_-protein under HF [56], suggesting that S616 may occur under chronic α_1_-AR stimulation, but the time course for this phosphorylation is different from that for S637 (see also Section 4.5 and Section 4.7). Therefore, further investigations are required to identify the upstream kinases for S616 phosphorylation under acute α_1_-AR stimulation and the onset of the activation of this signaling cascades under chronic α_1_-AR stimulation.

### 4.5. Role of DLP1 Phosphorylation at S616 in Mitochondrial Morphology and Function during Adrenergic Stimulation

The next question is whether S616 phosphorylation of DLP1 by chronic β-AR and/or α_1_-AR modulates mitochondrial morphology and function in CMs. Recently, Wang and Sheu’s group proposed that DLP1 has both canonical (i.e., regulation of mitochondrial fission) and non-canonical roles (e.g., regulation of transient mPTP opening and OXPHOS) in adult CMs [25]. Therefore, it is still unclear whether this phosphorylation promotes canonical functions of DLP1 (i.e., increased mitochondrial fission via S616 phosphorylation), and/or increases non-canonical functions of DLP1 (i.e., acceleration of transient mPTP opening independently of fission activity) (Figure 3B) [57]. Further studies may be needed to quantitatively assess whether the canonical and non-canonical functions of DLP1 are linked or occur independently of each other and how S616 phosphorylation regulates both functions in CMs under chronic β-AR stimulation. Lastly, since the upstream kinases for S616 phosphorylation under α_1_-AR stimulation have not yet been determined (see Section 4.4) and the changes in the level of S637 phosphorylation is likely more prominent than S616 phosphorylation under chronic α_1_-AR stimulation (see the following Section 4.7), further study will be necessary to investigate the relative contribution of the two phosphorylation sites to modulate mitochondrial fission under α_1_-AR stimulation in CMs (see also Section 4.8).

### 4.6. Phosphorylation of DLP1 at S637 by β-Adrenoceptor Signaling

Another phosphorylation site detectable under AR stimulation in CMs is S637. The sequences around the S637 match the consensus motif of PKA. Cribbs and Strack first showed that acute β-AR stimulation by an injection of Iso to the animals and exercise (forced swimming) promotes S637 phosphorylation in DLP1 in the heart in vivo, assuming that this phosphorylation is via PKA based on the observation from non-cardiac cells (Figure 3B) [58]. Next, Wang’s group confirmed using isolated adult CMs that 12-hr incubation with Iso exhibits increased S637 phosphorylation by PKA (but not via CaMKII) [57]. Moreover, they also found that this increased S637 phosphorylation disappears after 2-week Iso infusion in vivo [57]. These observations can be partly attributed to (1) the activation of Ca^2+^-dependent phosphatase CaN in the later phase of chronic β-AR stimulation (Figure 1 and Figure 3B) [191] and/or (2) the activation of PP2A via β_2_-AR-G_i_ pathway [192] (Figure 1 and Figure 3B), which may dephosphorylate DLP1 at S637 [58,60,153,154]. Thus, the phosphorylation levels of S637 may vary time-dependently during β-AR stimulation and HF development in vivo.

### 4.7. Phosphorylation of DLP1 at S637 by α_1_-Adrenoceptor Signaling

In addition to PKA, the sequences around the S637 also match to the consensus motif of PKD. Indeed, our group recently showed in neonatal CMs that 30-min treatment of an α_1_-AR agonist Phe in combination with a β-AR antagonist propranolol promotes S637 phosphorylation in DLP1 via PKD leading to increased association of DLP1 to the OMM [56]. Moreover, we also found a significant increase in S637 phosphorylation in ventricular tissues from mice with the overexpression of the constitutively active G_q_-protein concomitant with chronic PKD activation under HF [56], suggesting that S637 phosphorylation occurs in vivo via PKD activation during the course of acute and chronic α_1_-AR stimulation. However, Pennanen and his colleagues showed that long-term stimulation (24–48 h) with high concentrations of norepinephrine (which stimulates simultaneously both α_1_-AR and β-AR) causes dephosphorylation of S637 in DLP1 via a CaN-dependent mechanism [60], implying that AR subtypes may have cross-talk signaling that regulates the phosphorylation levels of DLP1 at S637. 

### 4.8. Role of DLP1 Phosphorylation at S637 on Mitochondrial Morphology and Function during Adrenergic Stimulation: Interplay of S637 and S616 Phosphorylation 

As briefly mentioned in Section 3, the functional consequences of S637 phosphorylation for the regulation of mitochondrial morphology are still highly controversial in both non-cardiac and cardiac cells. Our recent publication shows that PKD-mediated phosphorylation at S637 under G_q_PCR stimulation including α_1_-AR promoted mitochondrial fragmentation, confirmed by the overexpression of a S637A mutant (Figure 3B). Our results are consistent with reports of CaMKIα- and ROCK1-dependent DLP1 phosphorylation in non-cardiac cells [148,149], but these results show the opposite effect of what the majority of groups have reported [58,145,146,147,152,153,154]. This discrepancy may be partly due to the involvement of different upstream molecules in different cell types or in different AR-subtype stimulations in CMs, that may regulate not only the phosphorylation levels of S637 in DLP1, but also impact the PTMs of other site(s) within DLP1 or other mitochondrial fission/fusion machinery proteins. For instance, we assessed β-AR- and PKA-mediated changes in DLP1 phosphorylation, its translocation to the OMM, and mitochondrial morphology in our previous study [56]. We found that β-AR stimulation activates PKA and increases DLP1 phosphorylation at S637, which promotes DLP1 translocation to mitochondria in H9c2 cells, similar to the case of PKD-dependent DLP1 phosphorylation at S637 [56]. However, β-AR stimulation did not show any significant changes in mitochondrial morphology. To understand the molecular mechanisms producing this discrepancy between the effect of PKA and PKD on mitochondrial fission in H9c2 cells, we further assessed the changes in the phosphorylation levels of DLP1 at S616 before and after α_1_-AR stimulation using a dephospho-mimetic mutant of DLP1, DLP1-S637A. We found that α_1_-AR stimulation significantly decreases S616 phosphorylation in cells overexpressing a DLP1-S637A mutant. These data sets indicate that phosphorylation at one site within DLP1 may prevent or initiate the phosphorylation or dephosphorylation at the other sites by changing the access of other kinases or phosphatases, since S616 and S637 may be proximal within the 3D structure of DLP1 [100]. Indeed, in neurological diseases, the DLP1 S616/S637 phosphorylation ratio dictates the severity of pathogenesis [193,194]. Another speculation is that the basal S637 phosphorylation state is critical for maintaining S616 basal phosphorylation, indicating that a priming phosphorylation at S637 is required for S616 phosphorylation. The use of DLP1 mutants with double mutations at S616 and S637 [144] in CMs may provide us with more detailed mechanistic insights for the relative roles of these two sites in mitochondrial fission. In summary, it is likely that S637 phosphorylation by AR signaling participates in the mechanism for mitochondrial fragmentation under the chronic AR stimulation in the CMs. Future studies using DLP1 double mutants in CMs are needed to delineate whether the mechanisms for the S637 and S616 phosphorylation of DLP1 are linked to (or independent of) each other under AR stimulation, and if so, how the phosphorylation levels of these two sites cooperatively regulate the overall activity of DLP1 in CMs. These further studies would also provide us with mechanistic insights of the critical difference in the role of S637 phosphorylation on mitochondrial morphology/dynamics between non-cardiac and cardiac cells. 

### 4.9. Role of Adrenergic Signaling-Mediated DLP1 Phosphorylation in the Development of Heart Failure 

The remaining questions are (1) how does the status of S616 and/or S637 phosphorylation influence mitochondrial function in CMs under AR stimulation and (2) whether AR-mediated PTMs of DLP1 participate in the process of HF development. As discussed above, short-term stimulation of both β-AR and α_1_-AR mediates S637 phosphorylation [56,57] (Figure 3A) and this phosphorylation (at least by α_1_-AR-PKD signaling) is likely to enhance mitochondrial fission in CMs (Figure 3B). Both β- and α_1_-AR stimulations can sub-acutely increase respiration to match ATP production for this acute additional energy demand [56,71,195] and this effect is possible by the following three separated mechanisms:
(1)Increased [Ca^2+^]_m_ stimulates the TCA cycle (see also Section 2) [40].(2)Accumulation of DLP1 to the OMM and the subsequent enhancement of mitochondrial fission by PTMs of DLP1 increase the efficiency of the electron transport chain (ETC) by affecting ultrastructural and spatial organization of the respiratory chain and ATP synthase, which stimulates forward electron flow through the ETC and ATP production [196].(3)Fragmented mitochondria by PTMs of DLP1 can sustain higher [Ca^2+^]_m_ levels without propagation of Ca^2+^ to the neighboring mitochondria [197], which may also activate the TCA cycle to increase ATP generation.

This increased respiration by mitochondrial fission in CMs may be initially beneficial for the failing heart to boost its ATP production and maintain the ability to pump blood. Under well-coupled conditions, stimulation of forward flow by itself preferentially oxidizes the ETC and thereby lowers ROS levels [198]. However, increased electron flow at the ETC may oxidize the reduced forms of nicotinamide adenine dinucleotide and nicotinamide adenine dinucleotide phosphate (NADH and NADPH, respectively), thereby gradually depleting the anti-oxidative capacity [199]. Indeed, 24–48-h treatment of high-dose norepinephrine decreases oxygen consumption concomitant with lowering cellular ATP levels and increasing mROS levels in cultured neonatal CMs [60]. One possibility is that mitochondrial fragmentation via S637 phosphorylation of DLP1 under AR stimulation itself may structurally limit the size of the matrix cavity, which decreases the amount of anti-oxidative enzymes in the matrix. Though S637 phosphorylation may start to decrease due to the increased activity of phosphatases [60] during chronic AR stimulation, CaMKII-dependent S616 phosphorylation increases time-dependently, which increases the probability of mPTP opening [57]. Indeed, it is frequently observed that chronic AR stimulation not only causes mitochondrial fragmentation, but also increases the number of swollen mitochondria with structurally damaged mitochondrial cristae in isolated CMs and/or in vivo animal hearts, indicating enhanced mPTP opening and or OMM permeability under chronic AR stimulations [56,84,200]. Thus, DLP1 phosphorylation may be involved in upstream signaling for mPTP opening and excessive mROS production under chronic AR stimulation, which increases CM apoptosis followed by cardiac fibrosis, ROS-dependent HF signaling activation, and ROS-dependent arrhythmogenic events during HF. Finally, since DLP1 stimulates tBid-induced Bax oligomerization at OMM and promotes cytochrome c release in non-cardiac cells [201], further studies are also required for investigating whether DLP1 can promote OMM permeability via mPTP-independent pathway in CMs under chronic AR stimulations.

### 4.10. Other Post-Translational Modifications of DLP1 by Adrenergic Signaling

In addition to S616 and S637 phosphorylation, several phosphorylation candidate sites for glycogen synthase kinase 3β (GSK3β) within the structure of DLP1 have been reported [120,159], but the functional relevance of these sites have not yet been tested in CMs (Figure 2 and Table 1). GSK3β activity may be modulated under chronic AR stimulation (Figure 1), but the importance of this kinase activity for regulating mitochondrial function is well described in ischemic/reperfusion (I/R) injury [202] rather than non-ischemic HF induced by pressure overload or neurohumoral injury where chronic AR stimulation exists. In addition to phosphorylation, SUMOylation [127] and O-GlcNAcylation of DLP1 [129] are reported in CMs. SUMOylation of DLP1 promotes its association with the OMM followed by increased CM apoptosis [127], but it is still unknown whether this pathway is activated under AR stimulation during HF. The O-GlcNAcylation of DLP1 associated with hyperglycemia may be involved in diabetes-induced mitochondrial dysfunction such as diabetic cardiomyopathy [129], rather than non-ischemic HF. Currently, there are no reports regarding the role of DLP1 ubiquitination and S-nitrosylation in CMs (see Table 1), though emerging evidence has revealed that these PTM mechanisms are important for cardiac remodeling under chronic β-AR stimulation in the heart [203,204]. Specifically, DLP1 ubiquitination-mediated DLP1 degradation may have an important role for the development of HF since CM-specific DLP1-knockout mice exhibit a detrimental cardiac phenotype by the disturbances of mitochondrial autophagy (i.e., mitophagy) group [205,206]. Sadoshima’s group proposed using the CM-specific DLP1-knockout mice that endogenous expression of DLP1, translocation of DLP1 to the mitochondria, and DLP1-dependent mitophagy are required for protecting the heart against pressure overload-induced mitochondrial dysfunction and HF [205,206]. DLP1 expression in the heart indeed decreases under pressure overload [205], but the detailed mechanism for causing decreased DLP1 is not still clear in this model. Furthermore, this group also reported that mitophagy during myocardial ischemia is mediated via Rab9-associated autophagosomes and S616 phosphorylation of Drp1 by Rip1 [143]. Further studies are needed to investigate whether cardiac AR signaling has a potential to modulate DLP1-mediated mitophagy under pathological conditions.

## 5. Summary and Concluding Remarks

As summarized in this review, there has been substantial progress in understanding the detailed molecular mechanisms of how PTMs of DLP1, especially phosphorylation of DLP1 at S616 and S637, modulate important mitochondrial functions such as mitochondrial fission, bioenergetics, ROS generation, mPTP activity, and mitophagy using non-cardiac cells. Moreover, thanks to the recent studies using cultured CM-model cells, adult CMs, and cardiac-specific DLP1 knockout mouse lines, the biological importance of DLP1 in cardiac physiology and its contribution to cardiac pathologies have emerged. Phosphorylation of DLP1 by the downstream kinases of AR signaling likely contributes to the development of cardiac dysfunction such as non-ischemic HF. However, it is still up for debate whether altered levels of PTMs in DLP1 are a reflection of overall impairment of cellular function at the end-stage of the disease, or DLP1 PTM-mediated mitochondrial dysfunction serves as one of the pathogeneses of cardiac disease. Furthermore, though this review does not focus on other types of HF such as I/R injury, myocardial infarction, and diabetic cardiomyopathy (see also Section 4.10), it is also possible that other upstream signaling (e.g., ROCK [141], GSK3β [120,159] and CaN [151] under I/R injury and myocardial infarction)-mediated phosphorylation and other types of PTMs may have significant roles for the development of HF. Approaches such as quantitative measurements of the mitochondrial morphology/dynamics, bioenergetics, mROS production, and mPTP activity during the time course of AR stimulation in cultured CM-model cells, adult CMs, and in vivo animal models may provide further information regarding the cardiac-specific regulation of DLP1 function by AR signaling. Lastly, in addition to cardiac-specific DLP1 knockout animals, the development of knock-in mouse lines carrying DLP1 mutations that prevent or enhance its PTM effects (e.g., dephospho- or phospho-mimetic mutant of DLP1) will help resolve the remaining questions in the future.

## Figures and Tables

**Figure 1 antioxidants-07-00195-f001:**
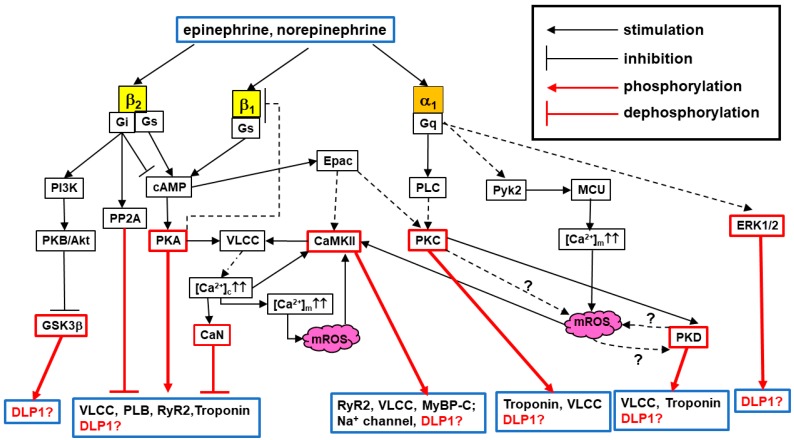
Overview of adrenergic signaling pathways in cardiomyocytes: Schematic diagram of cardiac AR subtypes and their downstream signaling. α_1_-AR consists of three highly homologous subtypes, including α_1A_-, α_1B_-, and α_1D_-AR, but we only show the downstream pathways from the main subtype in CMs, α_1A_-AR, in this diagram. Major Ca^2+^ handling proteins are also shown as the substrates for the downstream kinases of ARs. Dot lines indicate the pathways with multiple steps that are abbreviated in this scheme. Dot lines with “?” indicate the pathways with less evidence from publications, but highly promising pathways. PI3K, phosphoinositide 3-kinase; PLC, phospholipase C; PKB, protein kinase B; Pyk2, Protein tyrosine kinase 2; Epac, exchange factor directly activated by cAMP; MyBP-C, myosin-binding protein C; PP2A, Protein phosphatase 2; mROS, mitochondrial ROS; GSK3β, glycogen synthase kinase 3β; CaN, calcineurin.

**Figure 2 antioxidants-07-00195-f002:**
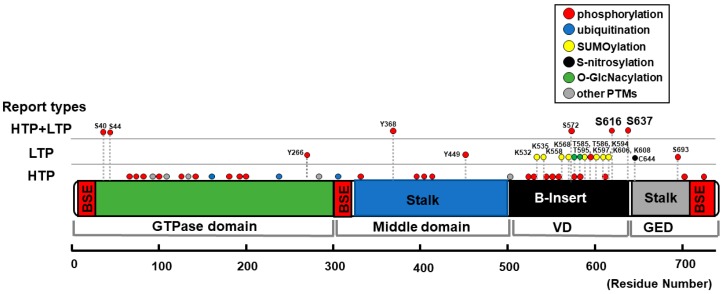
Schematic diagram of DLP1 structure and distribution of PTMs. Structure-based domain architecture of human DLP1 isoform 1 is depicted by the modified information of DLP1 isoform 2 from Fröhlich et al. [100]. If the publication record(s) is (are) only from proteomics and mass spectrometry data (“high-throughput papers” [HTP]), these sites are shown as HTP. The HTP information was collected from PhosphoSitePlus [102]. If the function of the specific site is investigated by methods other than high-throughput mass spectrometry (e.g., mutagenesis), then these records are categorized as “low-throughput papers” (LTP). If both low- and high-throughput records exist, then these records are shown as HTP + LTP. All the reports from LTP are summarized in Table 1. “Other PTMs” include acetylation, methylation, and proteolytic cleavage. If the information is from rat and mouse DLP1, site numbers are shown as the orthologous sites in human. BSE, bundle signaling element; GED, GTPase effector domain; VD, variable domain.

**Figure 3 antioxidants-07-00195-f003:**
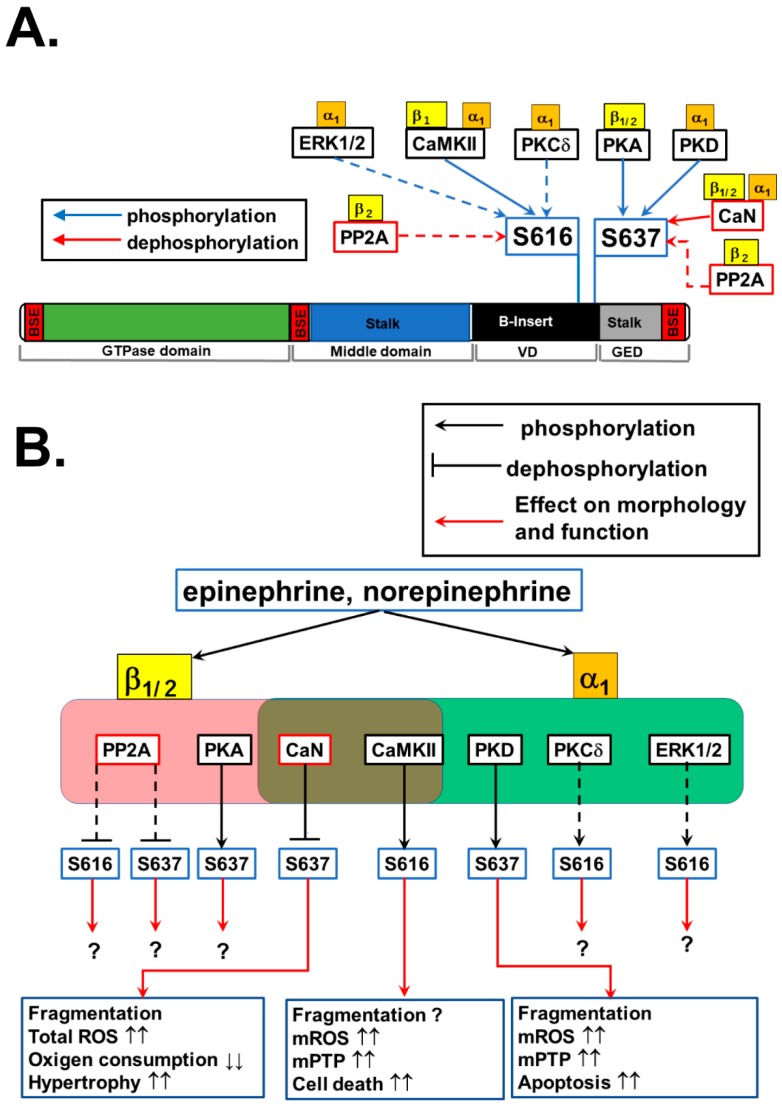
Phosphorylation of DLP1 and their effects on mitochondrial morphology and mitochondrial and cellular functions in cardiomyocytes: (**A**) DLP1 phosphorylation at S616 and S637 and their potential upstream signaling under AR stimulation. The upstream AR subtype(s) of each enzyme is(are) also shown. Dotted lines indicate the pathways reported in non-cardiac cells, but not yet tested in CMs. PP2A, Protein phosphatase 2; CaN, calcineurin. (**B**) Schematic diagram of DLP1 phosphorylation at S616 and S637 by AR signaling and their effects on mitochondrial morphology as well as mitochondrial and cellular functions in CMs. Dotted lines indicate the pathways reported in non-cardiac cells. If the functional effect of the signaling pathway has not yet been tested in CMs, question mark “?” is shown. The effects of CaN, CaMKII, and PKD are extracted from the following publications [56,57,60].

**Table 1 antioxidants-07-00195-t001:** Post-translational modifications of DLP1 and their effects on mitochondrial morphology.

Position	Type of Modification	Type of Reports	Upstream Molecules	Effect on Mitochondrial Morphology	Detected/Tested in CMs?
S40 S44	phosphorylation	HTP + LTP	GSK3β [120]	phosphorylation → fragmentation [120]	No (primary cultured hippocampus neurons [120])
Y266 Y368, Y449	phosphorylation	LTP only or HTP + LTP	c-Abl [121]	phosphorylation → fragmentation [121]	No (neuron-specific c-Abl knockout mice [121])
K532 K535 K558 K568 K594 K597 K606 K608	SUMOylation	LTP	Ubc9 [122] MAPL [123,124] SENP5 [125,126,127]	SUMOylation → fragmentation [122,123,124] deSUMOylation → elongation [125] → enlarged and swollen mitochondria [127] → fragmentation [126]	Yes [127]
S572	phosphorylation	HTP + LTP	CDK5 [128]	phosphorylation → fragmentation [128] dephosphorylation → elongation [128]	No (cerebellar granule neurons [128])
T585 T586	O-GlcNAcylation	LTP	N.D. (O-GlcNAc-transferase?)	O-GlcNAcylation → fragmentation [129]	Yes [129]
T595	phosphorylation	LTP	LRRK2-G2019S [130]	phosphorylation → fragmentation [130] dephosphorylation → elongation [130]	No (HeLa cells, HEK293 cells, human fibroblasts [130])
S616	phosphorylation	HTP + LTP	CDK1/cyclin B [131,132,133,134] CDK5 [135] ERK1/2 [103,136,137,138]. PKCδ [133,139] CaMKII [57,140] ROCK [141] PP2A [142] Rip1 [143]	phosphorylation → fragmentation [103,133,136,137,138,142] → elongation [135] dephosphorylation → elongation [133,139] → fragmentation [135] S616D → increased fission event rate [140] → no significant changes [144] → elongation [135] S616A → elongation [131] → fragmentation [56,135] → no significant changes [132,137,139,141,144]	Yes [56,57,141,143] (See also foot notes for ^1^ [57], ^2^ [103], ^3^ [133], and ^4^ [56])
S637	phosphorylation	HTP + LTP	PKA [58,145,146,147] CaMKIα [148] ROCK1 [149] PKD [56] Pim-1 [150] CaN [58,60,133,144,151,152] PP2A [153,154]	phosphorylation → fragmentation [56,148,149] → elongation [58,145,146,147,152,153,154] dephosphorylation → fragmentation [60,133,151,153,154] S637D → elongation [58,144,145,152] → fragmentation [148] S637A → fragmentation [58,144,152] → elongation [148] → no significant changes [56]	Yes [56,58,60,150] (see also foot notes for ^3^ [133], and ^5^ [150])
C644	S-Nitrosylation	LTP	NO [155,156,157]	S-Nitrosylation → fragmentation [156,157,158] → no significant changes [155]	No (Neurons [156,157,158], human brain tissue [155])
S693	phosphorylation	HTP + LTP	GSK3β [159]	phosphorylation → elongation [159] dephosphorylation → fragmentation [159]	No (HeLa cells, HEK293 cells [159])
N.D.	ubiquitination	HTP + LTP	MARCH5 [160,161,162,163] Parkin [164,165]	ubiquitination → degradation of DLP1, followed by elongation [160,161,164,165] → recruitment of DLP1 to the OMM followed by fragmentation [162,163]	No (HeLa cells [160,162,164], COS7 cells [161], RGC5 cells [163], SH-SY5Y cells [164,165]. (See also foot note for ^6^ [166])

HTP, high-throughput papers (see also Figure 2); LTP, Low-throughput papers (see also Figure 2); LRRK2, Leucine-rich repeat kinase 2; c-Abl, abelson murine leukemia viral oncogene homolog 1; CDK, Cyclin-dependent kinase; N.D., not determined; Pim-1, Proto-oncogene serine/threonine-protein kinase; GSK3β, glycogen synthase kinase 3β; CaN, calcineurin; Rho-associated protein kinase (ROCK). ^1^ Xu et al. detected DLP1 phosphorylation at S616 and S637 in CMs, but the effect of CaMKII on mitochondrial morphology was analyzed in H9C2 cardiac myoblasts [57]. ^2^ Yu et al. detected phosphorylation of DLP1 at S616 using an in vitro kinase assay and mitochondrial morphological changes were analyzed in H9c2 cells [103]. ^3^ Zaja et al. used the HL-1 CM line derived from the AT-1 mouse atrial myocyte tumor lineage for detecting DLP1 phosphorylation at S616 and S637 [133]. ^4^ Jhun et al. detected DLP1 phosphorylation at S616 in H9c2 cells, and heart trusses, but the effect of S616 phosphorylation on mitochondrial morphology was analyzed in H9C2 cells [56]. ^5^ Din et al. showed that Pim-1 decreases DLP1 expression and increases S637 phosphorylation, which results in mitochondrial elongation [150]. ^6^ Song et al. showed that cardiac-specific parkin ablation moderated hyper-mitophagy in DLP1-deficient hearts in vivo, but the phenotype of single deletion of Parkin in the heart was not documented [166].

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
