# Peer review of "Adrenergic Regulation of Drp1-Driven Mitochondrial Fission in Cardiac Physio-Pathology"

_antioxidants, 2018, doi:10.3390/antiox7120195_

Round 1
Reviewer 1 Report
This is an extensive well written review.
Still to make it more relevant for human pathologies
Suggest to mention the human the consequences DNM1L mutations on pathology and mitochondrial function
Elaborate also on peroxisomal function
Author Response
We thank the reviewer 1 for the constructive comments and suggestions for strengthening our manuscript. The significant changes we have made in this revised manuscript are highlighted in yellow. Following the editor’s suggestion, we kept all the changes in our manuscript using the world file “Track changes” function. Please use “Simple Markup Mode” to review this revised manuscript.
Dr. Michael Cypress had significant contribution for revising our manuscript and joined as a coauthor for our revised manuscript (see page 1 in our revised manuscript).
Revised manuscript has received professional service for the extensive editing of English language style following the reviewer 3’s suggestion.
We attach also the PDF file including the answers to all the reviewers' comments for your reference.
Q1: This is an extensive well written review.
Still to make it more relevant for human pathologies
Suggest to mention the human the consequences DNM1L mutations on pathology and mitochondrial function
A: We added the information and references regarding human DNM1L mutations following the reviewer’s suggestions (see page 10 line 25 to 31 in the revised manuscript).
Q2: Elaborate also on peroxisomal function
A: We added the descriptions regarding the DLP1 function on peroxisome following the reviewer’s suggestions (see page 5 line 7 to 10, page 5 line 32 to 34, and page 6 line 14 to 23 in the revised manuscript).

Reviewer 2 Report
The review manuscript is very well written and well organized, and will help the field significantly. Following are some minor comments:
1. Since the focus is on Drp1 (DLP1), the title should have 'Drp1 driven mitochondrial fission'. Drp1 is a more commonly used nomenclature, using DLP1 may reduce the citation of the review.
2. In Pg 2, 2nd para, relevant work from David Chan's lab should be included as well. They created a mice where this can be studied in an in vivo set up. Also, the work from Balaban group is focused on communication through junctions and does not demonstrate there is continuity in the matrix all along the conducting elements. A sentence to express that will communicate the idea more precisely.
3. Lines 43-51: the logic is not clear and over reaching.
4. PMIDs 20850011 and 25149858 should be discussed at relevant sections
5. The discussion on mitophagy is less adequate. It is not clear why any Drp1-s role in mitophagy should be called non-canonical.
6. A very important point is to cover the drug mdivi-1. Since mdivi-1 drug has been used by various labs to inhibit Drp1 (even in CMs), a small cautinary note about the reported off target effect of mdivi-1 should be included. The pronounced off target effect of Drp1 should discourage people from using it further. Also, any reported data on mdivi-1 that is not supported by Drp1 repression/inactivation should be considered a possible artifact due to the off-target effect.
Author Response
We thank the reviewer 2 for the constructive comments and suggestions for strengthening our manuscript. The significant changes we have made in this revised manuscript are highlighted in yellow. Following the editor’s suggestion, we kept all the changes in our manuscript using the world file “Track changes” function. Please use “Simple Markup Mode” to review this revised manuscript.
Dr. Michael Cypress had significant contribution for revising our manuscript and joined as a coauthor for our revised manuscript (see page 1 in our revised manuscript).
Revised manuscript has received professional service for the extensive editing of English language style following the reviewer 3’s suggestion.
We attach also the PDF file including the answers to all the reviewers' comments for your reference.
The review manuscript is very well written and well organized, and will help the field significantly. Following are some minor comments:
Q1: Since the focus is on Drp1 (DLP1), the title should have 'Drp1 driven mitochondrial fission'. Drp1 is a more commonly used nomenclature, using DLP1 may reduce the citation of the review.
A: We agree with the reviewer. We changed the title and revised our abstract using the term “Drp1”.
Q 2: In Pg 2, 2nd para, relevant work from David Chan's lab should be included as well. They created a mice where this can be studied in an in vivo set up. Also, the work from Balaban group is focused on communication through junctions and does not demonstrate there is continuity in the matrix all along the conducting elements. A sentence to express that will communicate the idea more precisely.
A: The reviewer is correct. We added the relevant work from David Chan's lab (see page 2, line 10 to 14 in the revised manuscript) and and also revised the description of the work from Balaban group following the reviewer’s suggestions (see page 2 line 17 to 21).
Q3: Lines 43-51: the logic is not clear and over reaching.
A: Reviewer is correct. We reduced the tone of our logic, simplified our idea, and rewrote this section (see page 4, line 45 to 51).
Q4: PMIDs 20850011 and 25149858 should be discussed at relevant sections
A: Following the reviewer’s suggestion, we added these publications with the descriptions in our revised manuscript (see page 6 line 2, page 6 line 8 to 9, page 15 line 20 to 23).
Q5: The discussion on mitophagy is less adequate. It is not clear why any Drp1-s role in mitophagy should be called non-canonical.
A: We removed “mitophagy” from the list of non-canonical function of DLP1 (see page 13 line 6). We also added further discussion of mitophagy following the reviewer’s suggestion (see page 15 line 42 to 50).
Q6: A very important point is to cover the drug mdivi-1. Since mdivi-1 drug has been used by various labs to inhibit Drp1 (even in CMs), a small cautionary note about the reported off target effect of mdivi-1 should be included. The pronounced off target effect of Drp1 should discourage people from using it further. Also, any reported data on mdivi-1 that is not supported by Drp1 repression/inactivation should be considered a possible artifact due to the off-target effect.
A: We agree with the reviewer’s comments. We added a discussion regarding the use of mdivi-1 and added a small cautionary note about the reported off target effect of mdivi-1 following the reviewer’s suggestions (see page 10 line 50 to page 11 line 9).

Reviewer 3 Report
Relationship between mitochondrial dynamics and pathology, especially for the cardiac dysfunction is an interesting topics and the manuscript of Jhun et al. is a well written comprehensive review on the topic. I really recommend the manuscript is suitable for publication in the journal, and only a comment may further strength their overview.
Drp1 function in mitochondrial fission process is no double about for the primary task. However, it is also known Drp1 acts for peroxisome division. Are there any cardiac physio-pathology regarding to the organelle?
Author Response
We thank the reviewer 3 for the constructive comments and suggestions for strengthening our manuscript. The significant changes we have made in this revised manuscript are highlighted in yellow. Following the editor’s suggestion, we kept all the changes in our manuscript using the world file “Track changes” function. Please use “Simple Markup Mode” to review this revised manuscript.
Dr. Michael Cypress had significant contribution for revising our manuscript and joined as a coauthor for our revised manuscript (see page 1 in our revised manuscript).
Revised manuscript has received professional service for the extensive editing of English language style following the reviewer 3’s suggestion.
We attach also the PDF file including the answers to all the reviewers' comments for your reference.
Reviewer 3
Relationship between mitochondrial dynamics and pathology, especially for the cardiac dysfunction is an interesting topics and the manuscript of Jhun et al. is a well written comprehensive review on the topic. I really recommend the manuscript is suitable for publication in the journal, and only a comment may further strength their overview.
Q1: Drp1 function in mitochondrial fission process is no double about for the primary task. However, it is also known Drp1 acts for peroxisome division. Are there any cardiac physio-pathology regarding to the organelle?
A: We added the description and updated information regarding DLP1, and peroxisomal morphology and function following the reviewer 1 and 3’ suggestions (see page 5 line 7 to 10, page 5 line 32 to 34, and page 6 line 14 to 23 in the revised manuscript).
